# Performance of methods for SARS-CoV-2 variant detection and abundance estimation within mixed population samples

Tunc Kayikcioglu[1,2], Jasmine Amirzadegan[1,3], Hugh Rand[1], Bereket Tesfaldet[1], Ruth E. Timme[4] and James B. Pettengill[1]

[1] Biostatistics and Bioinformatics Staff, Office of Analytics and Outreach, Center for Food Safety and Applied Nutrition, US Food and Drug Administration, College Park, MD, United States of America
[2] Joint Institute for Food Safety and Applied Nutrition, University of Maryland College Park, College Park, MD, United States of America
[3] Oak Ridge Institute for Science and Education, Oak Ridge, TN, United States of America
[4] Division of Microbiology, Office of Regulatory Science, Center for Food Safety and Applied Nutrition, United States Food and Drug Administration, College Park, MD, United States of America

Corresponding author
James B. Pettengill,
james.pettengill@fda.hhs.gov

## ABSTRACT

**Background**. The accurate identification of SARS-CoV-2 (SC2) variants and estimation of their abundance in mixed population samples (*e.g.*, air or wastewater) is imperative for successful surveillance of community level trends. Assessing the performance of SC2 variant composition estimators (VCEs) should improve our confidence in public health decision making. Here, we introduce a linear regression based VCE and compare its performance to four other VCEs: two re-purposed DNA sequence read classifiers (Kallisto and Kraken2), a maximum-likelihood based method (Lineage deComposition for Sars-Cov-2 pooled samples (LCS)), and a regression based method (Freyja).

**Methods**. We simulated DNA sequence datasets of known variant composition from both Illumina and Oxford Nanopore Technologies (ONT) platforms and assessed the performance of each VCE. We also evaluated VCEs performance using publicly available empirical wastewater samples collected for SC2 surveillance efforts. Bioinformatic analyses were performed with a custom NextFlow workflow (C-WAP, CFSAN Wastewater Analysis Pipeline). Relative root mean squared error (RRMSE) was used as a measure of performance with respect to the known abundance and concordance correlation coefficient (CCC) was used to measure agreement between pairs of estimators.

**Results**. Based on our results from simulated data, Kallisto was the most accurate estimator as it had the lowest RRMSE, followed by Freyja. Kallisto and Freyja had the most similar predictions, reflected by the highest CCC metrics. We also found that accuracy was platform and amplicon panel dependent. For example, the accuracy of Freyja was significantly higher with Illumina data compared to ONT data; performance of Kallisto was best with ARTICv4. However, when analyzing empirical data there was poor agreement among methods and variations in the number of variants detected (*e.g.*, Freyja ARTICv4 had a mean of 2.2 variants while Kallisto ARTICv4 had a mean of 10.1 variants).

**Conclusion**. This work provides an understanding of the differences in performance of a number of VCEs and how accurate they are in capturing the relative abundance of

SC2 variants within a mixed sample (*e.g.*, wastewater). Such information should help officials gauge the confidence they can have in such data for informing public health decisions.

# INTRODUCTION

SARS-CoV-2 (SC2) was declared a global pandemic by the World Health Organization (WHO) on 11 March 2020, and as of 16 September 2022, 13,134,400 clinical samples were sequenced globally and deposited into the GISAID SC2 database (*GISAID, 2022*). While providing a very accurate means for variant typing at an individual level, the financial burden of this individual-level clinical sequencing approach is high. There is also a sampling bias associated with clinical surveillance efforts. For example, due to different levels of access for different segments of the community and differences among groups in reporting rates to healthcare institutions. Wastewater-based epidemiological surveillance (WBS) (*Ramuta et al., 2022*; *Boogaerts et al., 2021*; *Sims & Kasprzyk-Hordern, 2020*) may be a cost-effective alternative or complimentary approach to wide-scale clinical surveillance that can measure disease agent prevalence among residents localized to a certain area, such as sewer shed or even a specific building. WBS has been shown to be a highly sensitive and specific method capable of detecting variants as well as providing the potential to detect the emerging ones (*Weidhaas et al., 2021*; *Crits-Christoph et al., 2021*; *Godinez et al., 2022*).

Apart from rare events such as co-infection or mutagenesis within the host, each clinical specimen has a single source and represents only one SC2 variant. In contrast, a wastewater sample is an unknown mixture of all infected individuals within the area, and can contain multiple lineages. This creates challenges for using WBS as an effective and accurate epidemiological tool to gauge community dynamics. There are issues with sample collection, concentration, and quantification of fragmented SC2 genomes existing at low viral loads. The subsequent targeted sequencing and library preparation to ensure efficient capture of the viral SC2 genomic content of the sample is also challenging (*Lu et al., 2020*; *Pulicharla, Kaur & Brar, 2021*; *Alhama et al., 2021*). In addition to these laboratory-based issues, there are those associated with the bioinformatic processing of sequence data to determine which variants are present and, particularly vexing, obtaining a reliable estimate of their relative abundance within the mixture (*Cao et al., 2021*; *Karthikeyan et al., 2022*).

Borrowing from the field of signal processing, in particular the principles associated with deconvolution, methods for SC2 variant composition estimation within mixed population samples have begun to emerge. For example, Freyja (*Karthikeyan et al., 2022*) calls single nucleotide polymorphisms of known SC2 variants based on the UShER (*Turakhia et al., 2021*) global phylogenetic reference tree. It then deconvolves relative abundances of the variants by fitting a weighted least absolute deviation based model and calculates weights from the read counts. Other methods, such as Kallisto (*Bray et al., 2016*), were originally designed for RNA-seq and metatranscriptomics, where the problem is analogous to that

presented by wastewater (*i.e.,* how to determine identity and abundance among a collection of fractionated transcripts). Kallisto uses de Bruijn graphs to pseudoalign k–mer classes to the sequences in a user provided reference database and provides final abundance estimates by optimizing a likelihood function. Another method, Lineage deComposition for SARS-CoV-2 pooled samples (LCS) (*Valieris et al., 2022*), uses a mixture model to provide a maximum-likelihood estimate of the relative abundances of different SC2 variants found in a sample based on a previously defined set of polymorphisms from a known set of variants. Taxonomic classifiers, such as Kraken2 (*Wood, Lu & Langmead, 2019*), notably increase the computational efficiency by using pre-indexed k-mer tables, and can be re-purposed to detect variants within a mixed sample by providing custom-curated reference databases (*e.g.*, a custom database of SC2 variants).

Here, we compare the performance of five variant composition estimators (VCEs). Two are recently published tools (Freyja and LCS), two others—Kraken2 and Kallisto— were not originally developed for processing SC2 sequence data but were optimized for that purpose here, and we developed a fifth VCE we call LINDEC that is linear deconvolution by least squares. We also introduce an *in silico* simulator of next-generation sequencing (NGS) test datasets of known variant composition and abundance along with a robust bioinformatics analysis pipeline. The pipeline was used to analyze the simulated data and empirical data collected through FDA's Center for Food Safety and Applied Nutrition's wastewater surveillance effort that leverages the genomic surveillance network for foodborne enteric pathogens, GenomeTrakr (*Allard et al., 2016*). Our primary objective was to determine which variant composition estimator has the lowest deviation from the simulated abundances. The empirical data was analyzed to confirm whether the performance and agreement among VCEs assessed *via* the simulated data is what is to be expected in a real-world application of those VCEs.

## MATERIALS AND METHODS

### Variant composition estimators

The VCE we developed, LINDEC, is available within our custom bioinformatics analysis pipeline C-WAP (CFSAN Wastewater Analysis Pipeline) and is based on linear deconvolution, which uses standard linear regression of a logical matrix of all mutations present or absent in a particular variant. In this representation, we pre-compile a list of all mutations present in all constellations reported in https://github.com/cov-lineages/constellations/tree/main/constellations/definitions. Let the total number of all mutations be $n$, which considers multiple mutation outcomes of the same genomic coordinate as distinct mutations. Thus, each lineage can be represented by an $n$-dimensional logical vector ($\mathbf{v}_i \in \{0, 1\}^n$) depending on whether a particular mutation in the pre-compiled mutation list is present or absent. The database itself then consists of an $n \times m$ dimensional matrix ($\mathbf{M} \in \{0, 1\}^{n \times m}$), where $m$ is the number of lineages included. A sample is processed by read alignment to reference sequence NC_045512.2, followed by pile-up generation, and variant calling against the reference sequence. The output is converted to an $n$-dimensional vector of rational numbers ($\mathbf{b} \in \mathbb{Q}^n$) and we do a least-squares fitting with Python's sci-kit

learn linear regression module (*Pedregosa et al., 2011*) to find the optimal coefficient vector **a** approximating the observed mutation, *i.e.,*

$$LINDEC = \text{argmin}_{\mathbf{a}} ||\mathbf{b} - \mathbf{Ma}||_2$$

with the constraint that $\mathbf{a} \in [0, 1] \cap \mathbb{Q}^m$ and the normalisation constraint $\mathbf{a}^T \mathbf{a} = 1$.

Another VCE, which we call ALLCB, employs both Kraken2 (*Wood, Lu & Langmead, 2019*), which is a taxonomic read classifier typically used for shotgun metagenomic data, and Bracken(Bayesian Reestimation of Abundance with KrakEN) (*Lu et al., 2017*). For this Kraken2/Bracken combined method, we initially included in our database the Wuhan reference sequence (NC_045512.2) and VOI/VOC lineage-representative assembly sequences published by US CDC (https://github.com/CDCgov/datasets-sars-cov-2/blob/master/datasets/sars-cov-2-voivoc.tsv). To capture additional variants, we also added lineages explicitly reported by CDC COVID Data Tracker (https://covid.cdc.gov/covid-data-tracker/#variant-proportions) by selecting three example assembly sequences obtained from GISAID database of each lineage; assemblies were selected based on completeness of sequence coverage, lowest ambiguous nucleotide counts (N), and chosen from three geographically well-separated areas when available. A list of included sequences is provided in Table 1.

To improve specificity (*i.e.,* the ability to differentiate two closely related lineages), we indexed this database with a longer k-mer length of 75 nucleotides (nts) instead of the publishers default of 35 nts. Reads that are shorter than the chosen k-mer length cannot be classified, and as a result, the read length of the short read platforms after primer trimming imposes an upper bound around this value in practice.

For Kallisto, a third VCE, we indexed the manually curated database described above using the default parameters. However, for the read classification we use the single read version with a mean read length of 300 nts and a spread of 50 nts. Kallisto can have multiple representatives per variant and in such instances we used the average of the estimated abundance across each representative to estimate the variant level abundance. Freyja and LCS represent the fourth and fifth VCEs and were used with their default configurations. For the LCS reference database, we used the pre-generated Pango designation marker table v1.2.124 provided by the developers.

## Simulations

We developed a custom workflow to simulate next-generation sequence data with known variant abundances (https://github.com/CFSAN-Biostatistics/ww_simulations). The workflow first generates random abundances for a user provided list of variant IDs. The second step involves generating amplicons from each genome using in_silico_PCR.pl (*Ozer, 2017*) based on a specific amplicon panel (*e.g.,* NEB's VarSkip Short multiplex PCR v1a). Using the *in silico* amplicons for each genome, the next step is to generate short read Illumina MiSeq data *via* ART v2.5.8 (*Huang et al., 2012*) or Oxford Nanopore Technology (ONT) sequence data *via* DeepSimulator v1.5 (*Li et al., 2018*). To model differences in abundances, a single amplicon file is generated within which each amplicon from each variant is represented by its expected abundance (*e.g.,* if Delta is expected to be 10% of a

**Table 1 Contents of Kraken2 and Kallisto DBs constructed for this study.** "Variant" refers to the GISAID-reported variant via Pangolin.

| Accession # | Variant | Accession # | Variant |
|---|---|---|---|
| NCBI NC_045512 | Wuhan-Hu | EPI_ISL_6810485 | BA.1 |
| EPI_ISL_1052966 | Alpha | EPI_ISL_6810487 | BA.1 |
| EPI_ISL_1519095 | Beta | EPI_ISL_6825397 | BA.1 |
| EPI_ISL_1365182 | Gamma | EPI_ISL_8679094 | BA.1 |
| EPI_ISL_836881 | Eta | EPI_ISL_9408266 | BA.1 |
| EPI_ISL_836839 | Eta | EPI_ISL_8881737 | BA.1 |
| EPI_ISL_802998 | Epsilon | EPI_ISL_8444273 | BA.1.1 |
| EPI_ISL_911639 | Epsilon | EPI_ISL_8929305 | BA.1.1 |
| EPI_ISL_803016 | Epsilon | EPI_ISL_9504608 | BA.1.1 |
| EPI_ISL_1615877 | Delta | EPI_ISL_8770510 | BA.2 |
| EPI_ISL_1631836 | Delta | EPI_ISL_8923845 | BA.2 |
| EPI_ISL_855171 | Iota | EPI_ISL_9449617 | BA.2 |
| EPI_ISL_1625962 | Iota | EPI_ISL_8975532 | BA.3 |
| EPI_ISL_1631305 | Kappa | EPI_ISL_8975536 | BA.3 |
| EPI_ISL_1719127 | Kappa | EPI_ISL_9431889 | BA.3 |

**Notes.**

The GenBank reference sequence, used as wt of this study, can be accessed under: https://www.ncbi.nlm.nih.gov/nuccore/NC_045512/. Other data originating from GISAID entries can be accessed under https://www.epicov.org/epi3/frontend#3eb7b8 with a valid user account.

sample and Alpha is expected to be 90% each will have 10 and 90 replicates per amplicon, respectively). Finally, a set of FASTQ files are generated within which reads from each genome are proportional to the abundances provided.

The validity of the simulations for the purpose of evaluating the VCEs was assessed by comparing the frequency of each mutation in the simulated (*i.e.,* observed) data to what was expected (*i.e.,* the known mutations in each lineage and the known abundance of that lineage in the simulation). We computed Lin's concordance correlation coefficients (*Lin, 1989*) to measure the agreement between the observed and expected frequencies across the 100 simulations. The values of Lin's CCC range from 0.8397 to 0.9487 with mean $0.8857 \pm 0.0249$ which are quite high and provides support for the simulations accurately representing the expected mutations and thus the signal being correctly measured by the VCEs.

Five different simulated datasets were generated for each of the two sequencing platforms (Illumina and ONT) and for some of the possible amplicon panels (Table 2). All simulations included five variants: Alpha (EPI_ISL_1052966), Beta (EPI_ISL_1519095), Delta (EPI_ISL_1615877), Epsilon (EPI_ISL_803016), and Omicron (EPI_ISL_14439177) (Fig. 1A), which differed in the number of *in silico* amplicons that could be generated for each of the three primer panels (Table 3). Five variants were chosen as that reflects the diversity that may be found at a given point in time within the human population (Fig. 1B). Some simulations had fewer than five variants because, due to chance, the abundance was set to 0 for certain variants. We also acknowledge there are situations where fewer or more variants are circulating within a community and that additional amplicon panels and

**Table 2  Details on the simulations including sequence type, amplicon panels, and parameters.**

| Sequencing platform | Simulator | Amplicon panels | Parameters[1] |
|---|---|---|---|
| Illumina | Art | ARTICv4 | L75/F150 |
| Illumina | Art | ARTICv4 | L150/F300 |
| Illumina | Art | QIAseq DIRECT | L75/F150 |
| Illumina | Art | QIAseq DIRECT | L150/F300 |
| Illumina | Art | NEB VSS v1a | L75/F150 |
| Illumina | Art | NEB VSS v1a | L150/F300 |
| Oxford Nanopore | DeepSimulator1.5 | NEB VSS v1a | L600 |
| Oxford Nanopore | DeepSimulator1.5 | NEB VSS v1a | L600 |

**Notes.**
[1] For Illumina, L = read length; F = fragment length with a standard deviation of 10 nts. For ONT, L = read length.

**Table 3  Amplicon panel metrics per variant.**

| | Amplicon panel | | |
|---|---|---|---|
| | ARTICv4 | NEB VSS v1a | QIAseq DIRECT |
| Amplicon length | | | |
| Min | 384 | 529 | 156 |
| Max | 420 | 604 | 305 |
| No. amplicons per variant | | | |
| Alpha | 92 | 68 | 394 |
| Beta | 94 | 70 | 399 |
| Delta | 95 | 72 | 389 |
| Epsilon | 93 | 70 | 388 |
| Omicron | 89 | 66 | 392 |

variants could have been considered. However, we believe the results presented here are worthy and provide critical information to understanding that not all variant estimators perform equally and the role that amplicon panel, sequencing technology, and differences in the genomes among variants have in explaining differences in performance.

## Empirical data

To further evaluate the performance of the variant estimators, we ran them across a suite of empirical data (Table S1). While the true variant composition is unknown for these samples, empirical data analyses may reveal insights in variability of VCE method predictions. This further provides basis in interpreting the *in silico* results, suggesting whether the expectations from the simulated data results are likely to be true when the VCE methods are employed in practice. All empirical data were sequenced using the Illumina platform and each of the three amplicon panels analyzed in the simulations are represented (Table 4). Due to limited availability of samples sequenced on the Oxford Nanopore platform, no ONT empirical data is included.

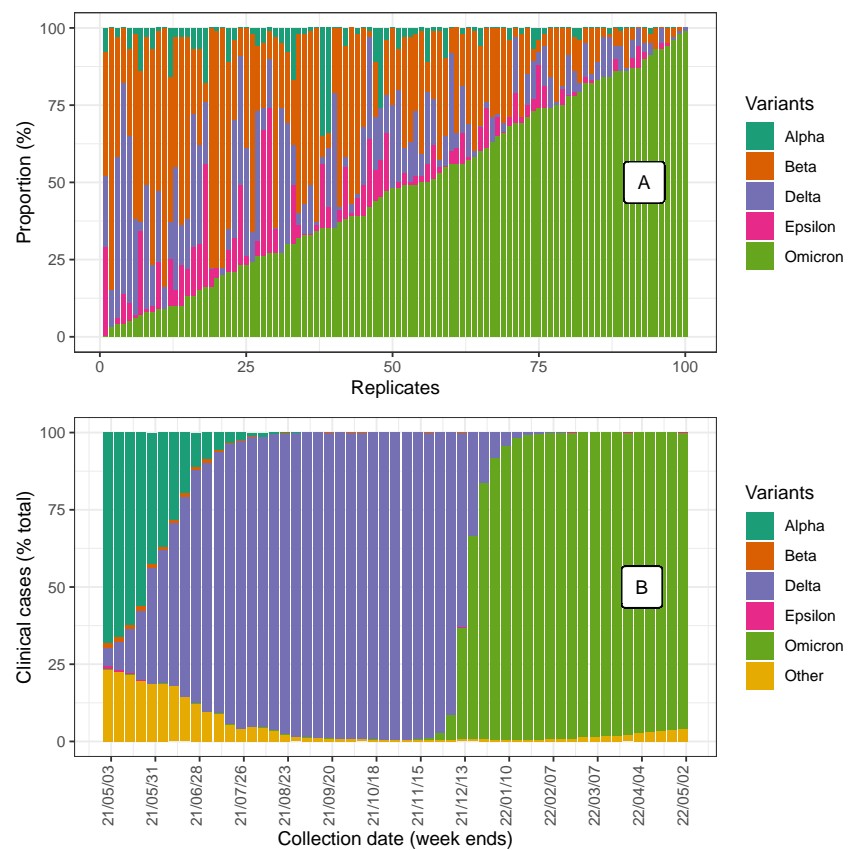

**Figure 1** (A) The variant composition of the 100 simulated datasets (arbitrarily sorted by increasing proportion of Omicron). (B) A snapshot of SC2 variants' prevalence for the past year, estimated out of GISAID metadata accessed on 08 June 2022 (EPI_SET ID: EPI_SET_220928co, https://doi.org/10.55876/gis8.220928co).

**Table 4** Characteristics of the Illumina empirical data.

| Amplicon panel | #SRRs | Library amplicon size (bp) | NCBI bioproject accession |
|---|---|---|---|
| ARTICv4 | 39 | 400 | PRJNA765612 |
| NEB VSS v1a | 15 | 250-490, 560, or UNK[1] | PRJNA767800, PRJNA757447 |
| QIAseq DIRECT | 69 | 300 | PRJNA757447 |

**Notes.**
[1] UNK denotes sequence data with missing information as to the library amplicon size.

## Bioinformatics analysis *via* C-WAP

Here, we introduce the CFSAN Wastewater Analysis Pipeline (C-WAP, https://github.com/CFSAN-Biostatistics/C-WAP), which performs bioinformatic analyses of SC2 sequence data from wastewater samples (or other mixed population samples) generated on Illumina, Oxford Nanopore, and PacBio sequencing platforms. It is a collection of python and shell scripts and open-source software dependencies that are managed *via* Conda and NextFlow (*Di Tommaso et al., 2017*). It quantifies SC2 present in NGS data *via* a mapping-based approach, characterizes the composition of the sample *via* a suite of variant estimator
methods (*i.e.,* those evaluated here), and generates HTML and PDF summary reports. C-WAP with default settings was used to analyze both the simulated and empirical datasets described here. This means that for Illumina read alignment we use Bowtie2 with default settings, whereas for ONT reads we use Minimap2 optimised for nanopore ("-x map-ont" option). Reads are not merged and for primer and quality trimming, we use iVar with default settings, except that for ONT we lowered the minimum quality threshold to 1 rather than the default value of 20. All detected mutations were included in the lineage abundance estimations without an explicit depth or frequency cutoff, as long as iVar indicates statistical significance.

Additionally, the estimators vary regarding the precision in the lineage predictions. To account for this, we parse their output *via* a custom Python3 script to retain the sub-lineage level detail only up to the lineages defined by WHO. For example, if the raw output predicts the composition to be 40% B.1.351.1 and 60% B.1.351.3, the results are reported as 100% B.1.351, which is defined as the Beta variant by WHO. We also present results from the analysis of the empirical data to demonstrate without parsing the results to account for how the VCEs differ in the lineage and sub-lineage level of reporting.

**Measure of variant composition estimation accuracy**

For the simulated data, we used relative root mean squared error (RRMSE) (*Xia et al., 2011*) to evaluate the estimation accuracy of each abundance estimation method. Let $n-1$ be the number of known variants in a simulated data, $(a_1, a_2, \ldots, a_{n-1})$ be the actual relative abundance of the known variants, and $(\hat{a}_1, \hat{a}_2, \ldots, \hat{a}_n)$ be the estimated relative abundance of the variants where $\hat{a}_n$ represents the relative abundance of a collection of variants that are absent from the simulated data. Then, RRMSE is given by

$$RRMSE = \sqrt{\frac{1}{n-1}\sum_{k=1}^{n-1}\left(\frac{\hat{a}_k - a_k}{a_k}\right)^2}.$$

Multivariable linear models were used to compare the five estimation methods and evaluate the effects of sequencing platforms, amplicon panels, and read/fragment lengths on the performance of the estimation methods. We also considered the same models but adjusting for the effect of genome sequencing coverage. Given that the distribution of RRMSE is right skewed, the error measures are log-transformed, and thus, we perform all our inferences on the expectations of the log-transformed error measures rather than the error measures themselves. Tukey-Kramer method was used to adjust for multiple testing in comparing each pairwise combination between the methods. We used an alpha level of 0.05 for all statistical tests.

We used concordance correlation coefficient (CCC) with Euclidean distance to evaluate the levels of pairwise agreement among the VCEs for both the simulated and empirical datasets. It was adapted by *Cui et al. (2021)* from Lin's concordance correlation coefficient for agreement studies with microbiome data. Let $\mathbf{a}_i = (\hat{a}_{i1}, \hat{a}_{i2}, \ldots, \hat{a}_{in})$ and $\mathbf{b}_i = (\hat{b}_{i1}, \hat{b}_{i2}, \ldots, \hat{b}_{in})$ denote a pair of estimated relative abundances of variants from data

$i$ ($i = 1, 2, \ldots, m$) based on two methods A and B, respectively. Then, the sample estimate of CCC between A and B is given by

$$CCC = \frac{\frac{2}{m}\sum_{i=1}^{m}\langle \mathbf{a}_i - \bar{\mathbf{a}}, \mathbf{b}_i - \bar{\mathbf{b}}\rangle}{\|\bar{\mathbf{a}} - \bar{\mathbf{b}}\|^2 + \frac{1}{m}\sum_{i=1}^{m}\left(\|\mathbf{a}_i - \bar{\mathbf{a}}\|^2 + \|\mathbf{b}_i - \bar{\mathbf{b}}\|^2\right)}$$

where $\bar{\mathbf{a}}$ is a vector that represents the component-wise average of $\mathbf{a}_i$'s, $\langle \mathbf{x}, \mathbf{y} \rangle$ is the inner product of vectors $\mathbf{x}$ and $\mathbf{y}$, and $\|\mathbf{x}\|$ is the norm of the vector $\mathbf{x}$. The values of CCC are between $-1$ and $1$, where $-1$ indicates a perfect disagreement, while $1$ indicates a perfect agreement. We used a bootstrap method of sample size 5000 to build a 95% confidence interval for each CCC estimate. For the empirical wastewater samples, we used Bayes-Laplace Bayesian-multiplicative replacement method to impute zero relative abundances (*Palarea-Albaladejo & Martn-Fernndez, 2015*) on the entire dataset before computing CCC values.

## RESULTS AND DISCUSSION

### Simulated data and the effect of sequencing platform

We first evaluated the performance of the five abundance estimation methods on simulated data from two sequencing platforms: Illumina and Oxford Nanopore. The results show that Kallisto was the best performing method as it had the least average RRMSE of $0.20 \pm 0.127$ in Illumina data and $0.18 \pm 0.140$ in Oxford Nanopore data followed by Freyja $0.31 \pm 0.194$ in Illumina and $0.39 \pm 0.159$ in Nanopore (Table 5). The CCC between Kallisto and the actual relative abundances (ACTUAL) is 0.9852 (95% CI [0.9820–0.9878]) and with that of Freyja is 0.9943 (95% CI [0.9926–0.9956]) in Illumina; for Oxford Nanopore data the CCCs are 0.9955 (95% CI [0.9946–0.9962]) with actual and 0.9799 (95% CI [0.9748–0.9840]) with Freyja (Fig. 2). The CCCs between Freyja and the actual are 0.9925 (95% CI [0.9910–0.9938]) and 0.9702 (95% CI [0.9629–0.9762]) in Illumina and Oxford Nanopore, respectively. These results show that the relative abundance estimates of variants using Kallisto and Freyja are very close to the actual relative abundances, and there is a very close agreement between the two methods in their estimations.

We compared the performance of the estimation methods using a two-way ANOVA with interaction on the log-transformed RRMSE estimates, where the two fixed factors are sequencing platform as two levels, and estimation methods as five different levels. The results indicate that there is a statistically significant interaction ($F(4, 1490) = 10.92$, $p < 0.0001$) between sequencing platforms and among abundance estimation methods on the accuracy of abundance estimation. Thus, the accuracy of some estimation methods depend on the platform used. Further analyses show that, except of Freyja and ALLCB, the accuracy of all VCE methods do not significantly differ across platforms used. Freyja has a statistically significant ($t(1490) = -4.73$, $p = 0.0001$) better accuracy on Illumina compared to Oxford Nanopore, while ALLCB has a statistically significant ($t(1490) = -3.26$, $p = 0.0380$) better accuracy on Oxford Nanopore compared to Illumina. Kallisto has better accuracy on Oxford Nanopore compared to Illumina, though the difference is not

**Table 5** **Summary (mean ± SD) of RRMSE by platform and method.** The sample size for Illumina is 200, while that of Oxford Nanopore is 100.

| Method | Illumina | Nanopore |
|---|---|---|
| ALLCB | 1.65 ± 1.42 | 1.25 ± 1.04 |
| FREYJA | 0.31 ± 0.19 | 0.39 ± 0.16 |
| KALLISTO | 0.20 ± 0.13 | 0.18 ± 0.14 |
| LCS | 1.22 ± 2.53 | 0.90 ± 1.02 |
| LINDEC | 1.02 ± 0.80 | 1.04 ± 0.74 |

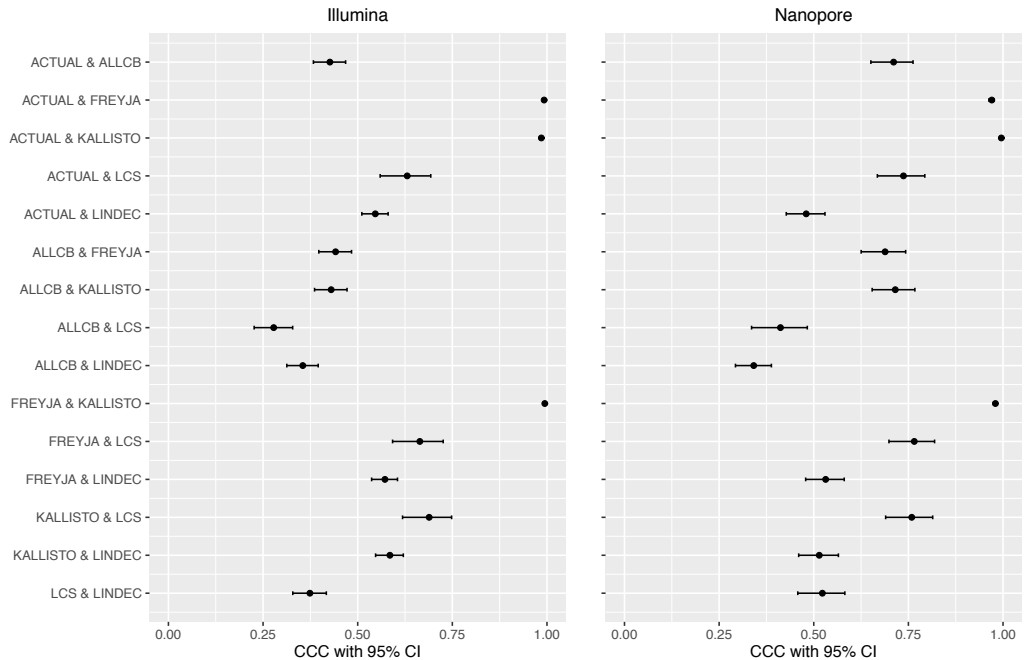

**Figure 2** **Concordance correlation coefficient (CCC) with bootstrap 95% lower and upper confidence interval (CI) limits between methods including actual abundances for both platforms used.** ACTUAL = known relative abundance of variants in simulated data.

statistically significant ($t(1490) = -2.86$, $p = 0.1179$). Overall, Kallisto and Freyja are significantly the most accurate estimation methods regardless of the platform used.

Looking at genome sequencing coverage, on average Illumina has higher percent of genome with missing coverage of $2.40 \pm 0.63\%$ than that of Oxford Nanopore which is $1.74 \pm 0.31\%$. Due to a lack of primer binding sites, most of the missing positions are at the 5′ ($\sim$ 300 nt) and 3′ ($\sim$ 100 nt) ends of the reference genome. The above conclusions still hold after controlling for the effect of genome sequencing coverage in the models. An exception is ALLCB which has a better accuracy with Oxford Nanopore compared to Illumina, but is no longer statistically significant ($t(1489) = -1.43$, $p = 0.9178$).
**Table 6   Summary of RRMSE (Mean ± SD) by Amplicon Panel and Read/Fragment Length for Illumina simulations.** Lower RRMSE means better performance and smaller deviation from the known relative abundance. L75F150 = read length of 75 and fragment size of 150; L150F300 = read length of 150 and fragment size of 300.

| Method | Read length | ARTICv4 | NEB VSS v1a | QIAseq DIRECT |
|---|---|---|---|---|
| ALLCB | L150F300 | 1.38 ± 1.034 | 1.73 ± 1.452 | 1.65 ± 1.334 |
| ALLCB | L75F150 | 1.80 ± 1.812 | 1.57 ± 1.381 | 1.44 ± 1.258 |
| FREYJA | L150F300 | 0.28 ± 0.194 | 0.30 ± 0.200 | 0.37 ± 0.167 |
| FREYJA | L75F150 | 0.31 ± 0.180 | 0.31 ± 0.189 | 0.63 ± 0.979 |
| KALLISTO | L150F300 | 0.20 ± 0.142 | 0.22 ± 0.153 | 0.27 ± 0.148 |
| KALLISTO | L75F150 | 0.18 ± 0.089 | 0.18 ± 0.092 | 0.52 ± 0.876 |
| LCS | L150F300 | 0.50 ± 0.181 | 0.44 ± 0.163 | 0.39 ± 0.130 |
| LCS | L75F150 | 0.59 ± 0.081 | 2.00 ± 3.407 | 0.65 ± 0.314 |
| LINDEC | L150F300 | 0.96 ± 0.700 | 1.02 ± 0.797 | 1.49 ± 1.445 |
| LINDEC | L75F150 | 0.92 ± 0.652 | 1.02 ± 0.811 | 1.55 ± 1.535 |

## Simulated data and the effects of amplicon panel and read length

Next, we evaluated the performance of the five abundance estimation methods on combinations of three amplicon panels: ARTICv4, QIAseq DIRECT, and NEB VSS v1a, and two read/fragment lengths: read length of 75 and fragment size of 150 (L75F150) and read length of 150 and fragment size of 300 (L150F300) for Illumina simulations. ALLCB consistently underestimates the variants Delta and Omicron, and overestimates the variants Alpha, Beta and Epsilon (Fig. S1). LCS underestimates the variants Delta and Epsilon, while LINDEC consistently underestimates the variants Beta and Omicron. Compared to other VCEs, both LCS and LINDEC have higher misclassification rates of variants present in the simulation datasets to variants absent from the datasets.

The accuracy measures show that Kallisto has the lowest average RRMSE within each combination ranging from $0.18 \pm 0.089$ to $0.52 \pm 0.876$; Freyja was the second lowest with RRMSE values ranging from $0.28 \pm 0.194$ to $0.63 \pm 0.979$ (Table 6). The CCC between Kallisto and the actual ranges from lowest 0.7714 (95% CI [0.6483–0.8552]) for read length 75 in QIAseq DIRECT to highest 0.9976 (95% CI [0.9971–0.9979]) for read length 150 in NEB VSS v1a (Table S2). The CCC between Freyja and the actual one ranges from lowest 0.7817 (95% CI [0.6579–0.8643]) for read length 75 in QIAseq DIRECT to highest 0.9971 (95% CI [0.9965–0.9977]) for read length 150 in NEB VSS v1a. For longer reads, LCS has higher CCCs with the actual one regardless of the amplicon panel used and the highest value is 0.9408 (95% CI [0.9234–0.9543]) for QIAseq DIRECT. The CCC between Kallisto and Freyja ranges from lowest 0.9736 (95% CI [0.9659–0.9796]) in QIAseq DIRECT to highest 0.9981 (95% CI [0.9977–0.9985]) in NEB VSS v1a for read length 150.

To compare the performance of the VCEs and evaluate the effects of amplicon panel and read/fragment lengths used on the performance of the methods, we fit a three-way ANOVA with interaction on the log-transformed RRMSE estimates. The results show that there is a statistically significant three-way interaction ($F(10, 2970) = 5.52$, $p < 0.0001$) indicating the accuracy of certain estimation methods may depend on the amplicon panel and the read/fragment length combination used. For example, Kallisto is a statistically significant

most accurate estimation method, followed by Freyja in almost all combinations of amplicon panel and read lengths used. An exception is the read length 150 in QIAseq DIRECT amplicon panel, where Freyja is more accurate than LCS, though this is not statistically significant ($t(2970) = -1.44$, $p = 0.9999$). Kallisto is a statistically significantly more accurate method compared to Freyja for read length 75 in ARTICv4 ($t(2970) = -4.85$, $p = 0.0005$) and NEB VSS v1a ($t(2970) = -4.54$, $p = 0.0022$), and for read length 150 in QIAseq DIRECT ($t(2970) = -3.86$, $p = 0.0348$). Most methods work accurately in longer read length though most are not statistically significant. Kallisto has a statistically significant better accuracy for longer read in QIAseq DIRECT ($t(2970) = -4.60$, $p = 0.0017$). LCS also performs better in both QIAseq DIRECT ($t(2970) = -5.64$, $p < 0.0001$) and NEB VSS v1a ($t(2970) = -10.78$, $p < 0.0001$). Kallisto and Freyja have better accuracy in both ARTICv4 and NEB VSS v1a than in QIAseq DIRECT, however Kallisto's accuracies in both ARTICv4 and NEB VSS v1a are not statistically significant compared to QIAseq DIRECT for read length 150 ($t(2970) = -3.73$, $p = 0.0543$; $t(2970) = -3.19$, $p = 0.2553$).

On average, QIAseq DIRECT has lower percent of genome with missing coverage of $1.23 \pm 0.35\%$ than that of ARTICv4 and NEB VSS v1a which are $2.43 \pm 0.88\%$ and $2.40 \pm 0.63\%$, respectively. The differences in genome sequencing coverage are negligible across read length for each amplicon panel. The above conclusions still hold after analyzing the data adjusting the model for the effect of genome sequencing coverage, except Kallisto's accuracy across amplicon panels. Kallisto's accuracy in both ARTICv4 and NEB VSS v1a turn out to be statistically significant better than in QIAseq DIRECT for read length 150 ($t(2970) = -5.87$, $p < 0.0001$; $t(2970) = -5.28$, $p < 0.0001$).

### Empirical data

We applied the five abundance estimation methods on 123 empirical wastewater samples, all sequenced on the Illumina platform: 39 were sequenced using ARTICv4 amplicon panel, 15 using NEB VSS v1a, and 69 using QIAseq DIRECT. Those lab/physical samples were collected between September 2021 and February 2022; see NCBI for sample specific dates. Table 7 shows the number of SC2 variants identified across empirical samples. Among the methods, on average LCS identified the highest number of variants with an average of $21.1 \pm 3.46$ variants (ranging from 7 to 24) in ARTICv4, $21.9 \pm 2.25$ (from 16 to 24) in NEB VSS1a, and $24.0 \pm 0.12$ (from 23 to 24) in QIAseq DIRECT across wastewater samples. Freyja identified the fewest number of variants with an average of $2.2 \pm 0.45$ variants (ranging from 2 to 4) in ARTICv4, $2.1 \pm 0.25$ (from 2 to 3) in NEB VSS v1a, and $3.0 \pm 0.79$ (from 2 to 5) in QIAseq DIRECT across wastewater samples. See Fig. S2 for additional information on the abundance estimates of specifics variants for identified in experimental wastewater sample data.

Pairwise concordance correlation coefficients (CCC) between the methods for each amplicon panel are shown in the forest plot in Fig. 3. The highest CCC is between Kallisto and LINDEC 0.9912 (95% CI [0.9745–0.9970]) followed by that of ALLCB and Kallisto 0.7679 (95% CI [0.5285–0.8942]) both in NEB VSS v1a amplicon panel. In general, there is a higher level of agreement between the VCEs with NEB VSS v1a amplicon panel than with the other amplicon panels. Compared to the simulated data, the CCC values between

**Table 7  Number of SC2 variants identified by each method in each amplicon panel across empirical wastewater samples.**

| Method | ARTICv4 Mean ± SD | Range | NEB VSS v1a Mean ± SD | Range | QIAseq DIRECT Mean ± SD | Range |
|---|---|---|---|---|---|---|
| ALLCB | 4.7 ± 1.48 | (1, 7) | 8.9 ± 0.81 | (7, 10) | 8.9 ± 0.92 | (6, 10) |
| FREYJA | 2.2 ± 0.45 | (2, 4) | 2.1 ± 0.25 | (2, 3) | 3.0 ± 0.79 | (2, 5) |
| KALLISTO | 10.1 ± 2.43 | (3, 12) | 11.9 ± 0.25 | (11, 12) | 12.0 ± 0.00 | (12, 12) |
| LCS | 21.1 ± 3.46 | (7, 24) | 21.9 ± 2.25 | (16, 24) | 24.0 ± 0.12 | (23, 24) |
| LINDEC | 7.7 ± 4.01 | (1, 15) | 3.0 ± 2.53 | (1, 9) | 3.3 ± 3.54 | (1, 14) |

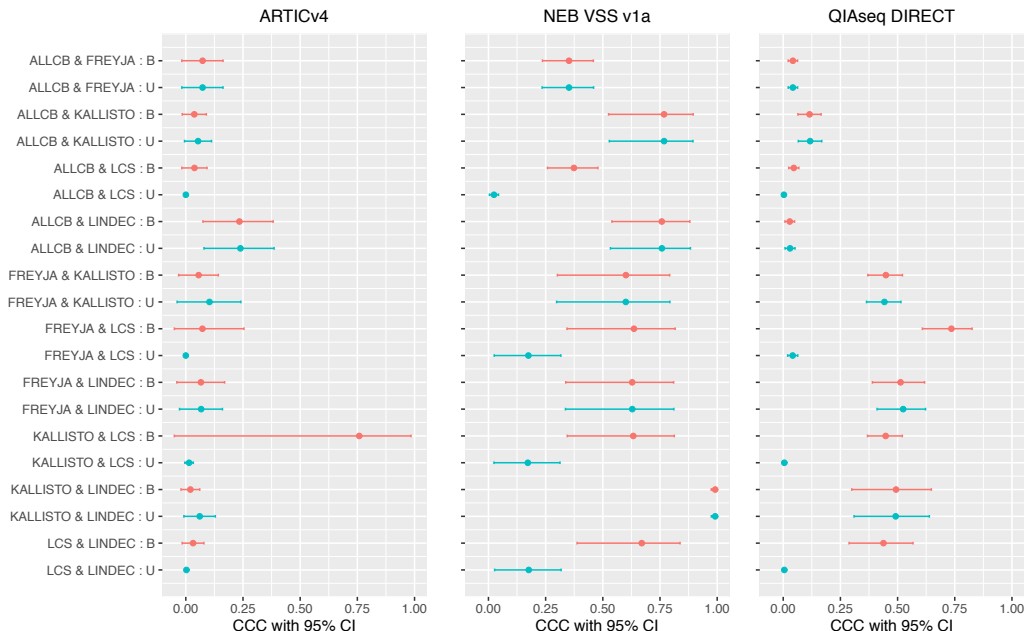

**Figure 3  Concordance correlation coefficient (CCC) with 95% lower and upper bootstrap confidence interval (CI) limits among methods on experimental wastewater sample data from all three amplicon panels.** U = CCCs from observed variants; B = CCCs after sub-lineages are binned together.

most of the methods in the empirical wastewater samples are much smaller. These smaller CCCs between VCEs for the empirical data can be explained in part by the fact that VCEs differ in their reference databases. Some are updated more frequently than others and, in this case, that meant that some (*e.g.*, LCS) included more sub-lineages that emerged within Omicron at the beginning of 2022 than others (*e.g.*, ALLCB, Freyja). This also explains in part the differences in the maximum number of variants detected by each VCE (Table 7).

To determine if these differences could be addressed by ensuring the VCEs were reporting similar taxonomic levels (*i.e.*, sub-lineages or not), we performed post-processing of the results and binned together sub-lineages (*e.g.*, BA1, BA2, and BA3 are binned as Omicron; AV1 and AV.1 as the same denoted *via* NA2) and recomputed the CCCs among the VCEs. Most CCCs between VCEs did not change much except the ones involving LCS. For example, the CCC between Kallisto and LCS increased from 0.0143 (95% CI

[−0.0042–0.0327]) to 0.7583 (95% CI [−0.4136–0.9845]) in ARTICv4, and from 0.1721 (95% CI [0.0242–0.3127]) to 0.6333 (95% CI [0.3447–0.8126]) in NEB VSS v1a. The CCC between Freyja and LCS increased from 0.1746 (95% CI [0.0251–0.3165]) to 0.6362 (95% CI [0.3435–0.8162]) in NEB VSS v1a, and from 0.0414 (95% CI [0.0192–0.0636]) to 0.7354 (95% CI [0.6084–0.8257]) in QIAseq DIRECT. The CCC between LINDEC and LCS increased from 0.1761 (95% CI [0.0270–0.3176]) to 0.6702 (95% CI [0.3876–0.8376]) in NEB VSS v1a. These results highlight that even when accounting for differences in the level of lineage and sub-lineage reporting and in the syntax/nomenclature of variants, that the VCEs still differ in their results. This has important implications given the results are dependent on the VCE being used.

## CONCLUSION

SC2 will inevitably evolve where new variants will arise and some form of monitoring for emerging variants will be an important component of public health efforts. Along with the evolution of new variants, sample kits, databases of variants, and detection methods will change. As the performance of these tools are likely dependent on the variant(s) in circulation at the time of the analysis, an ultimately conclusive assessment of the accuracy of the methods is elusive. However, the results presented here show the importance of the choice of laboratory methods (*i.e.,* sequencing technology and amplicon panel) and bioinformatic analytics (*i.e.,* variant estimators) in accurately detecting and quantifying variant composition within a mixed population sample such as wastewater. Although certain methods perform quite well with the simulated data and have a high CCC when compared to one another, the results for the experimental data illustrate that the picture of variants within a sample is quite dependent on the VCE being employed (even when controlling for differences in the ways and as what lineage/sub-lineage resolution the methods report results). This information and future evaluations of variant estimation methods are crucial to fulfill the promise often made that genomic epidemiology and the analysis of mixed population samples will enhance current and future responses to pandemics (*Knyazev et al., 2022*). Given that all methods evaluated here rely on a reference database of known SC2 variants against which reads are classified, a future challenge will be the development of variant estimators that reliably detect and characterize a novel variant not seen before and is absent from the reference databases.

## ACKNOWLEDGEMENTS

We gratefully acknowledge the CovidTrakr working groups at the Center for Food Safety and Applied Nutrition for work and discussion that facilitated this manuscript. We also acknowledge the support of CFSAN high performance computing engineers G. Engelbach, K. Konganti, and M. Hammond in the installation and maintenance of the analytic software we used and evaluated. We also acknowledge all data contributors for generating the genetic sequence and metadata and sharing *via* the GISAID Initiative, on which this research is partly based.

### Funding

Jasmine Amirzadegan's participation was supported by an appointment to the Research Participation Program at the U.S. Food and Drug Administration administered by the Oak Ridge Institute for Science and Education through an interagency agreement between the U.S. Department of Energy and the U.S. Food and Drug Administration. Tunc Kayikcioglu received financial support from Joint Institute for Food Safety and Applied Nutrition (JIFSAN), University of Maryland as part of financial assistance award U01FD001418 funded by the Food and Drug Administration (FDA) of the U.S. Department of Health and Human Services (HHS). The funders had no role in study design, data collection and analysis, decision to publish, or preparation of the manuscript.

### Grant Disclosures

The following grant information was disclosed by the authors:
U.S. Food and Drug Administration administered by the Oak Ridge Institute for Science and Education through an interagency agreement between the U.S. Department of Energy and the U.S. Food and Drug Administration.
Joint Institute for Food Safety and Applied Nutrition (JIFSAN).
Food and Drug Administration (FDA) of the U.S. Department of Health and Human Services (HHS): U01FD001418.

### Competing Interests

The authors declare there are no competing interests.

### Author Contributions

- Tunc Kayikcioglu conceived and designed the experiments, performed the experiments, analyzed the data, authored or reviewed drafts of the article, and approved the final draft.
- Jasmine Amirzadegan conceived and designed the experiments, performed the experiments, analyzed the data, authored or reviewed drafts of the article, and approved the final draft.
- Hugh Rand conceived and designed the experiments, authored or reviewed drafts of the article, and approved the final draft.
- Bereket Tesfaldet conceived and designed the experiments, analyzed the data, prepared figures and/or tables, authored or reviewed drafts of the article, and approved the final draft.
- Ruth E. Timme conceived and designed the experiments, authored or reviewed drafts of the article, and approved the final draft.
- James B. Pettengill conceived and designed the experiments, performed the experiments, analyzed the data, prepared figures and/or tables, authored or reviewed drafts of the article, and approved the final draft.

## Data Availability

The CFSAN Wastewater Analysis Pipeline is available at GitHub: https://github.com/CFSAN-Biostatistics/C-WAP.

The sequences are available at:

NCBI: NC045512, EPI_ISL_1052966, EPI_ISL_1519095, EPI_ISL_1365182, EPI_ISL_836881, EPI_ISL_836839, EPI_ISL_802998, EPI_ISL_911639, EPI_ISL_803016, EPI_ISL_1615877, EPI_ISL_1631836, EPI_ISL_855171, EPI_ISL_1625962, EPI_ISL_1631305, EPI_ISL_1719127, EPI_ISL_6810485, EPI_ISL_6810487, EPI_ISL_6825397, EPI_ISL_8679094, EPI_ISL_9408266, EPI_ISL_8881737, EPI_ISL_8444273, EPI_ISL_8929305, EPI_ISL_9504608, EPI_ISL_8770510, EPI_ISL_8923845, EPI_ISL_9449617, EPI_ISL_8975532, EPI_ISL_8975536, EPI_ISL_9431889

## Supplemental Information

Supplemental information for this article can be found online at http://dx.doi.org/10.7717/peerj.14596#supplemental-information.

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
