# Peer review of "Performance of methods for SARS-CoV-2 variant detection and abundance estimation within mixed population samples"

_PeerJ, doi:10.7717/peerj.14596_

## Round 0.1 · original submission · Minor Revisions

The 3 reviewers agree that your manuscript had lots of merits but requires some additional work. Please address all of their comments and suggestions before resubmitting.

Reviewer 1 ·

Basic reporting

The language used throughout the manuscript is clear, concise and professional. There are some typos and oversights (some listed in 4. additional comments) particularly in the materials and methods section and the text should be proofread carefully.

The authors provide sufficient background in a nicely written introduction citing major references and providing context for their efforts in benchmarking tools to distinguish SARS-CoV-2 variants in mixed samples derived from amplicon sequencing experiments.
The references should be re-checked as in at least one occasion a reference had been forgotten to be included (add "sci-kit learn citation" L95ff).

Figures and tables are relatively clear, however some minor improvements could be made:
- Figure1, a) and b) could have the same color scheme
- Table captions could be more elaborate in some cases. What does Null stand for in table3 amplicon size?
- Captions for supplementary figures are not provided

Even though references to raw data utilized in the study are included (NCBI project IDs for the empirical data) the information which samples had been selected from these bioprojects is missing and should be made available, e.g., as supplemental data/table. Additionally, in order to achieve reproducibility it would be required to add details on the simulated data (simulated reads or the input files and commands used to generate those reads with the scripts in https://github.com/CFSAN-Biostatistics/ww_simulations should be provided).
Furthermore, details on the databases (see 2. Experimental design) are missing.

Experimental design

Several aspects of the applied methods should be specified in greater detail:
LINDEC:
L95ff, speaking of SNPs instead of mutations would be more appropriate?
Why is a logical matrix used? In theory different mutations could at the same positions could be relevant, even though in praxis examples are quite rare.
Additional details should be provided for C-WAP, particularly on mapping and variant (SNP) calling. Are there cutoffs for mutations to be considered?
ALLCB:
setting the k-mer length to 111 supposedly has a relatively large impact and the authors mention that shorter reads reads cannot be classified. However, results are shown later on for data with read length 75. Are reads joined prior to classification?

Databases for ALLCB and kallisto:
The process of generating the databases should be outlined in greater detail and variants/genomes incorporated should be listed somewhere (supplemental data?). Linking to a github repository https://github.com/CDCgov/datasets-sars-cov-2 with (parts of?) the underlying data is not sufficient in my opinion.
In general, as the authors state themselves, differences in database generation likely cause much of the differences in performance. Without detailed description of the database generation process it is not clear whether the chosen methods are comparable.

L105, please specify if the same database is used as for ALLCB.

L151, please specify "custom python script", is it "getDisplayName.py"? The differences between the applied tools in their capabilities to resolve sub-lineages is not clear and thus the results on the empirical dataset remain challenging to interpret.
Also, it seems that CCCs for the empirical dataset were calculated on the raw prediction results (sub-lineages, figure S2) and not the "merged WHO lineages", L248ff? This was confusing to me because it differs to the results for the simulated data and the reasoning for choosing this approach was not clear.
The "extent to which raw results differ" (L255) between the computational methods could/should also be described elsewhere, but here it also affects the comparison.

Differences in sequencing depth and coverage can be critical in reliably calling and comparing mutations in mixed samples. For the simulated data it should be mentioned to what depth reads were generated and for the empirical data depth and coverage should be assessed to ensure the validity of the results.
Especially, for the comparison of amplification schemes, effects of coverage should be looked at.

Validity of the findings

In their manuscript the authors contribute to wastewater-based epidemiological surveillance methods by comparing and benchmarking 4 previously developed approaches and their own method to estimate the abundance of SARS-CoV-2 variants from mixed samples. For the comparison, they apply a simulated dataset of up to 5 variants and also assess empirical data derived from wastewater samples. Additionally, the authors provide an automated pipeline featuring their own deconvolution method.
The presented methods and findings could provide an important contribution to the field.

The experimental design is well-chosen, however key details are omitted from the methodological description (see 2. Experimental design) which makes it difficult to judge whether the shown results are valid, robust and/or meaningful.
While some raw data is provided, in its current form the study is not easily reproducible.

The description of the results on simulated data and applied statistical methods appear suitable, however the discussion of the results could be extended. In particular, reasons for the differences in accuracy of the tested approaches should be discussed.

Results on the empirical data are difficult to interpret and require additional context (expected variants, reporting on the bioinformatic data/processing). Conclusions drawn from the empirical part should be highlighted, for now it just seems trends in amplicon panels observed for the simulated data differ for the empirical data. The aim to use the empirical data "to confirm whether the performance assessed via the simulated data is what is to be expected in a real-world application of the VCEs" needs to discussed around L250.

Additional comments

Extra period L45
capitalization SARS-CoV L74
L77-L80 restructure sentence
L107 forthe
L106/L107 nt instead of nts as defined in L101
L129 extra that
L146 extra period and one missing

line numbers missing after L95 and L155, L162 (before formulas)

table1 parameters footnote?

Reviewer 2 ·

Basic reporting

The language used throughout is clear and unambiguous. Though there were several punctuation errors throughout and should be corrected. Some errors can be found in line 45, 146, and 147.

While I agree there may be sampling bias with monitoring sequencing clinical samples, especially as more people are using rapid tests. There are also some limitations to wastewater sequencing as a mechanism to monitor variants. In Smyth, et al. Tracking cryptic SARS-CoV-2 lineages detected in NYC wastewater, the authors detected cryptic variants and it was unclear where these variants originated. Are there any references that speak to the sensitivity and specificity of wastewater sequencing?

Supplemental Figure S1 has grey bar graphs in the background with no clear explanation of what those value represent. I recommend adding it to the legend to S1. Supplemental Figure S2 has displays a lot of variants making it difficult to discern some of the plots. I recommend, if possible, excluding some of the variants that are consistently in low abundance and explaining why they were left out of the figure.

Results are relevant to the hypothesis. The authors aimed to compare the tools used to estimate the abundance of the variants detected in mixed culture. They performed accuracy tests on simulated data and identified amplicon panel and read length that contributed to the most accurate results. Additionally, they analyzed the same tool.

Experimental design

This manuscript is a comparative analysis of the tools used to estimate the abundance of mix population of SARS-CoV-2 in a sample, which is extremely pertinent to studies involving environmental samples. I believe this fits in with the aims and scope of PeerJ.
The questions are clear and is relevant to the current landscape of community surveillance using environmental samples. Currently, there are not many papers published comparing tools and providing guidance as to which tools to use to perform variant abundance estimation. As the public health agencies are gathering more methods to help strengthen surveillance efforts, this research is an extremely helpful addition to the published works. The authors used adequate statistical tests and methods to explore and compare the bioinformatics tools used to analyze the amplicon based sequencing data of mix population samples.
I believe the method do describe sufficient detail for replication of analyzing simulated data and real data.

Validity of the findings

The author has effectively compared the tool using simulated data as well as real data. There were some clear differences in terms of the abundance estimation between the tools, but it is unclear which is most reliable for real data. I think this is really important, because considering wastewater could contain dirty sequences, cryptic sequences, or the sequencing methods could contain poor quality data, all of which could obscure the the abundance estimation. One thing I think would help strengthen the author's assessment is to perform some kind of verification that the which tools estimated the real data correctly. For instance, looking at the data used for the analyses, are there similar abundance of lineage defining mutations as compared to the tools used?

Another question I have is, were other features besides amplicon panel and read length considered for looking at the accuracy of the abundance estimation? In real data, especially with amplicon sequencing, there may be amplicon drop outs with regions of low depth of coverage and therefore incomplete genomes recovered. Do these have an impact on the abundance estimators?

Additional comments

I think this is a very important paper for public health surveillance effort and is up-to-date with the leading laboratory methods and computation tools that are currently used for sequencing and analyzing environmental samples. I think with some revisions made, this will help guide public health laboratories and bioinformaticians to select appropriate amplicon panel, read lengths, and tools for their SARS-CoV-2 monitoring efforts.

Reviewer 3 ·

Basic reporting

see attached

Experimental design

see attached

Validity of the findings

see attached

Additional comments

see attached

Annotated reviews are not available for download in order to protect the identity of reviewers who chose to remain anonymous.

---

## Round 0.2 · Minor Revisions

One of the reviewers still had a number of suggestions for changes. Please address these before resubmitting.

Reviewer 1 ·

Basic reporting

The authors addressed most comments and suggestions thoroughly.
However, several aspects have not been sufficiently addressed in the revised manuscript:
1. A list of reference genomes/lineages utilized for database construction is not provided.
L129ff (Line numbers refer to the track-changes version) is merely stating that some lineages/genomes have been added.
From the C-WAP repository it seems there are 2 distinct kraken2 databases with 27? genomes (majorcovidDB) and 66? genomes (allcovidDB), but I am not sure if these have been used in this study.

2. A list of empirical samples utilized in the study is not provided.

3. Raw data (predicted frequencies/relative abundances of lineages) on the comparisons used to calculate CCC values are not provided.

4. The language in the modified parts of the manuscript should be re-checked.

Points 2 and 3 could be addressed by providing a single supplemental table with the following information:
SRA_ID, bioprojectID, collection date/location, VCE utilized, lineage, predicted abundance
For the simulated data it would also be beneficial to provide a table with the theoretical abundances for each sample and the predicted abundances for each of the VCEs.

Experimental design

For the comparison of the empirical data very low abundances are frequently inferred (Figure S2). Are these points resulting from imputed 0 abundances?

For Table6, does the numbers of SC2 variants depend on a tool's tendency to predict low abundance lineages?

Validity of the findings

Given the data and information provided it is still not possible for me to assess the validity of the results.

L345ff
I appreciate the additional analysis done, however I have a hard time believing that the CCC values "didn't change much" after binning sub-lineages for the other methods given the extent that they changed for LCS.
It would be very possible to add this information (CCC values after processing lineages) to Figure 3, as well as providing source data (prediction results for the tools, see above).
The CCC values for LCS after binning lineages are comparable or higher in the empirical data than in the simulated data.

L355
The author's conclude that different VCEs differ in their results. I still think the average reader could benefit from an assessment on how the compared tools differ in their theoretical capabilities (as applied here), predicting (sub)-lineages or only major lineages.

Additional comments

L123 check capitalization of KrakEN
L263 check language
L309 check language
L340ff check language
Table6, missing comma (16 24)

The conclusions section remains quite vague and maybe more details could be added as well as recommendations on which methods/approaches work best and why.

Reviewer 2 ·

Basic reporting

The language used throughout is clear and unambiguous. Though, the authors did not respond to providing "references that speak to the sensitivity and specificity of wastewater sequencing". If there aren't any references available that's ok, but if there are, it would be beneficial to mention.

Experimental design

I believe the method do describe sufficient detail for replication of analyzing simulated data and real data.

Validity of the findings

I appreciate the authors including text to clarify abundance of lineages in simulated data and real world data. Also appreciate adding the acknowledgement that allele/amplicon drop outs may affect the variant detections.

---

## Round 0.3 · accepted · Accept

Please try to provide the requested table during the last review as a supplementary file or deposited in a public repository.

Reviewer 1 ·

Basic reporting

Basic reporting has improved to a point where I no longer think publication should be delayed any further. Theoretically, the analysis in the manuscript could be reproduced.

Raw data should not be provided as a table in the manuscript, but instead could be supplied as supplementary data files, ideally in machine readable format and, if necessary, through external repositories:
https://peerj.com/about/author-instructions/

Experimental design

-

Validity of the findings

-

Additional comments

-